# Molecular Mechanisms of L-Type Calcium Channel Dysregulation in Heart Failure

**DOI:** 10.3390/ijms26125738

**Published:** 2025-06-15

**Authors:** Arbab Khalid, Abu-Bakr Ahmed, Randeep Gill, Taha Shaikh, Joshua Khorsandi, Ali Kia

**Affiliations:** 1Department of Internal Medicine, Kirk Kerkorian School of Medicine, UNLV, Las Vegas, NV 89106, USA; arbab.khalid@unlv.edu (A.K.); randeep.gill@unlv.edu (R.G.); taha.shaikh@unlv.edu (T.S.); 2Kirk Kerkorian School of Medicine, UNLV, Las Vegas, NV 89106, USA; ahmeda15@unlv.nevada.edu (A.-B.A.); khorsand@unlv.nevada.edu (J.K.)

**Keywords:** L-type calcium channel (LTCC), Ca^2^⁺/calmodulin-dependent protein kinase II (CaMKII), heart failure, calcium handling, excitation–contraction coupling, T-tubule remodeling

## Abstract

The L-type calcium channels (LTCCs) function as the main entry points that convert myocyte membrane depolarization into calcium transients, which drive every heartbeat. There is increasing evidence to show that maladaptive remodeling of these channels is the cause of heart failure with reduced ejection fraction (HFrEF) and heart failure with preserved ejection fraction (HFpEF). Recent experimental, translational, and clinical studies have improved our understanding of the roles LTCC expression, micro-domain trafficking, and post-translational control have in disrupting excitation–contraction coupling, provoking arrhythmias, and shaping phenotype specific hemodynamic compromise. We performed a systematic search of the PubMed and Google Scholar databases (2015–2025, English) and critically evaluated 17 eligible publications in an effort to organize the expanding body of work. This review combines existing data about LTCC density and T-tubule architecture with β-adrenergic and Ca^2^⁺/calmodulin-dependent protein kinase II (CaMKII) signaling and downstream sarcoplasmic reticulum crosstalk to explain how HFrEF presents with contractile insufficiency and how HFpEF shows diastolic calcium overload and stiffening. Additionally, we highlight the emerging therapeutic strategies aimed at restoring calcium homeostasis such as CaMKII inhibitors, ryanodine receptor type 2 (RyR2) stabilizers, and selective LTCC modulators without compromising systolic reserve. The review establishes LTCC dysregulation as a single mechanism that causes myocardial dysfunction while remaining specific to each phenotype, thus offering clinicians and researchers a complete reference for current concepts and future precision therapy approaches in heart failure.

## 1. Introduction

L-type calcium channels (LTCCs) are important for carrying out cardiac excitation–contraction (EC) coupling. During every heartbeat, membrane depolarization results in the opening of LTCCs, which are mostly Ca_v1.2 in ventricular myocytes, allowing a small influx of Ca^2+^ that, in turn, triggers a much larger release of Ca^2+^ from the sarcoplasmic reticulum (SR), a process referred to as calcium-induced calcium release [1,2]. This LTCC-triggered Ca^2+^ signal elevates the cytosolic Ca^2+^ rapidly to activate myofilaments and produce contraction, and Ca^2+^ reuptakes into the SR then permits relaxation [1,2]. Because of this efficient coupling of electrical excitation to mechanical contraction by LTCCs, their activity is essential in maintaining cardiac output and rhythm.

In heart failure (HF), dysregulation of intracellular Ca^2+^ handling is a primary characteristic causing contractile dysfunction and arrhythmias [1,3,4]. Maladaptive changes of LTCC function or expression disrupt EC coupling to generate less effective contractions and vulnerability to arrhythmogenic afterdepolarizations. In total, HF is usually categorized into two phenotypes—heart failure with reduced ejection fraction (HFrEF) with dilated ventricles and systolic dysfunction, and heart failure with preserved ejection fraction (HFpEF) with near-normal systolic ejection but deranged diastolic relaxation [5]. Notably, HFrEF and HFpEF exhibit disparate remodeling processes and responses to treatments, suggesting that they have different molecular mechanisms [6,7]. It is important to understand how LTCC regulation is altered in both HF subtypes since LTCC dysfunction contributes to reduced contractility and arrhythmogenesis in HF [3,8].

This review was based on literature searches in the area using the PubMed (https://pubmed.ncbi.nlm.nih.gov, accessed on 15 April 2025) and Google Scholar (https://scholar.google.com, accessed on 15 April 2025) databases for published articles between 2015 and 2025, focusing on molecular alterations in LTCC regulation in HFrEF and HFpEF. We highlight how HFrEF is generally associated with reduced LTCC expression/function, altered post-translational modifications, and EC coupling “uncoupling”, whereas HFpEF tends to show preserved LTCC expression with maladaptive signaling changes and diastolic dysfunction. Emerging therapies for HF aim to normalize dysregulated Ca^2^⁺ handling by inhibiting CaMKII hyperactivity (using peptides like AIP, KN 93, or new small-molecule inhibitors [9]), stabilizing leaky ryanodine receptor type 2 (RyR2) channels (Rycals such as ARM210 to prevent diastolic Ca^2^⁺ leak [10]), and selectively modulating L-type Ca^2^⁺ currents (e.g., enhancing Rad–LTCC inhibitory interactions or blocking re-expressed Cav1.3 channels [11]).

## 2. Methods

A literature review was performed using the PubMed (https://pubmed.ncbi.nlm.nih.gov, accessed on 15 April 2025) and Google Scholar (https://scholar.google.com, accessed on 15 April 2025) databases. Keywords for this search included ““L-type calcium channel”, “excitation–contraction coupling”, “heart failure”, “HFrEF”, “HFpEF”, and “CaMKII”. Search results were filtered to English language only, covering the period from 2015 to 2025. Although this 10-year interval was extensive, the final selection of 17 publications was based on strict inclusion criteria focused on the specificity, quality, and relevance of the data to LTCC dysregulation in HFrEF and HFpEF. Article types included observational studies, retrospective studies, randomized controlled trials, systematic reviews, and meta-analyses.

## 3. Discussion

### 3.1. LTCC Dysregulation in HFrEF

Heart failure with reduced ejection fraction is classically associated with weakened EC coupling and depressed systolic Ca²⁺ transients [12,13,14]. The hallmark of HFrEF is the downregulation of LTCC expression and function in ventricular myocytes [15]. Studies in failing human hearts and animal models show that peak L-type Ca^2+^ current density is reduced, often by ~30–40% compared with normal, due to a loss of LTCCs in the membrane and/or altered channel gating [4,16]. This reduction in LTCCs contributes to smaller Ca^2+^ transients and contractile weakness in HFrEF [1,2,3,4]. Failing human ventricular myocytes exhibit significantly lower LTCC current and binding capacity, which can partly recover after mechanical unloading (LV assist device support) [16]. Likewise, chronic dilation and increased wall stress in HFrEF models lead to decreased t-tubular L-type Ca^2+^ current and uncoupling from SR Ca^2+^ release [4,17,18].

Along with a reduced number of channels, HFrEF is characterized by post-translational modifications—most notably phosphorylation of the LTCC α₁ and β subunits—as well as oxidative modifications and nitrosylation, which alter the channel gating and inactivation kinetics [19]. Early HF exposed to continuous β-adrenergic stimulation triggers the protein kinase A (PKA)-mediated phosphorylation of LTCC α_1 subunits as well as their accessory β subunits [4,8]. Acute protein kinase A phosphorylation strengthens L-type Ca^2+^ current but chronic hyperadrenergic conditions cause maladaptive phosphorylation leading to desensitization. HFrEF cells show hyperactivated CaMKII activity, which modulates LTCC subunit phosphorylation for activity regulation [8,20]. CaMKII phosphorylation promotes a higher open probability and prolonged openings of LTCC [8]. CaMKII facilitation (a CaMKII-mediated mechanism whereby repeated calcium influx increases the opening probability and current amplitude of L-type Ca^2^⁺ channels) of channels results in slower inactivation and total L-type Ca^2+^ current decreases [8]. These changes lead to early arrhythmogenic afterdepolarizations in failing cells [6]. HFrEF myocytes typically show reduced functional LTCC numbers but stronger protein kinase A and CaMKII phosphorylation along with incorrect channel gating [8].

A striking feature of HFrEF is EC coupling dysfunction (or “uncoupling”) due to structural remodeling. In healthy myocytes, LTCCs on transverse tubules (T-tubules) are tightly colocalized with ryanodine receptors (RyR2) on junctional SR, ensuring synchronized Ca^2+^ release. HFrEF is associated with the disruption and loss of T-tubules [4]. Studies have shown that failing ventricular cells develop a sparse, disorganized T-tubule network, yielding “orphaned” RyR2 that are physically distant from LTCC trigger sites (Figure 1) [18]. This leads to delayed, dyssynchronous Ca^2+^ release sparks and a lower probability of LTCC openings successfully activating RyR2 [4]. For example, in post-infarction HFrEF, Ca^2+^ spark probability and synchrony are markedly reduced, and Ca^2+^ release latency is prolonged, consistent with defective LTCC–RyR coupling [4]. T-tubule remodeling in HFrEF has been causally linked to these changes—elevated wall stress and stretch-mediated signaling in dilated hearts drive T-tubule loss and caveolin-3 dysfunction, thereby impairing LTCC microdomain signaling [5,17,18]. As a result, HFrEF cardiomyocytes exhibit depressed EC coupling gain (less SR Ca^2+^ released per unit LTCC current) and reduced contractile efficiency [4]. This uncoupling mechanism is a key contributor to systolic dysfunction in HFrEF [17].

In summary, HFrEF is characterized by a downregulation of LTCC expression/function and the structural uncoupling of LTCCs from RyR2 (Figure 1). Chronic neurohumoral activation leads to the altered phosphorylation of LTCCs (e.g., via CaMKII and protein kinase A), which can initially be compensatory but ultimately become maladaptive. These molecular and structural changes converge to suppress systolic Ca^2+^ release and contractility [3,4], while also creating an unstable Ca^2+^ environment that predisposes to arrhythmias (due to LTCC and RyR2-mediated triggered activity) [8]. HFrEF therapies like β-blockers and mechanical unloading likely benefit the heart in part by reversing some of these alterations by upregulating LTCC density, restoring T-tubule structure, and reducing hyperphosphorylation-driven channel dysfunction [5,16].

### 3.2. LTCC Dysregulation in HFpEF

LTCC regulation in HFpEF is categorically different. As opposed to HFrEF, systolic function is comparatively well-preserved in HFpEF (ejection fraction (EF) ≥ 50%). As a result of this, LTCC expression and Ca^2+^ influx are also comparatively well-preserved. A study undertaken by Kilfoil et al. indicates that LTCC density and whole-cell L-type Ca^2+^ current are preserved or even slightly increased in HFpEF models relative to normal [4]. In a hypertensive HFpEF rat model, baseline myocytes exhibited a larger L-type Ca^2+^ current amplitude and Ca^2+^ transient amplitude than the controls. The Ca^2^⁺ spark release probability per LTCC activation was increased, and the latency between LTCC opening and Ca^2^⁺ release was decreased in HFpEF cells, indicating more effective LTCC-RyR2 signaling compared with HFrEF. These unique adaptations enable HFpEF cardiomyocytes to preserve systolic Ca^2+^ release in the setting of diastolic dysfunction (Figure 1) [1].

Consistent with this, an optical imaging study in human HFpEF demonstrated that LTCC-mediated Ca^2+^ release was similarly well-preserved. Failing HFpEF patient myocytes lacked the prominent Ca^2+^ release desynchrony characteristic of HFrEF, and Ca^2+^ transient upstrokes in the failing myocytes were relatively coordinated [5]. The evidence in this case demonstrates that HFpEF myocytes exhibit normal LTCC numbers and function that allows the global EF to be normal or near normal.

The structure of transverse tubules remains nearly normal in HFpEF compared with the extensive T-tubule damage seen in HFrEF. Both human and animal hearts with HFpEF exhibited either unchanged or enhanced T-tubule density throughout their compacted hypertrophied hearts, as revealed by histological and imaging studies [5]. Frisk et al. reported that human HFpEF cardiomyocytes that experienced hypertrophic changes without developing severe dilation maintained high T-tubule density and regular organization, thus supporting effective LTCC–RyR coupling [5]. The structural continuity enables Ca^2+^ release to stay synchronized with LTCC trigger events in HFpEF. Normal T-tubule architecture in rat HFpEF models enables near-normal spark synchrony and Ca^2+^ release timing [4]. HFpEF shows no sign of the serious EC decoupling that characterizes HFrEF, instead, HFpEF cells typically exhibit efficient EC coupling at the baseline [4]. An increased spark trigger probability, together with higher synchrony, was observed in supra-normal coupling fidelity by one study that proposed this as a compensatory mechanism to counteract stiffening of the ventricular wall in HFpEF [4]. This adaptation enables stroke volume to be preserved when relaxation is compromised [4].

In healthy heart muscle cells, the transverse tubules or T-tubules form a well-organized network that ensures that the L-type calcium channels (LTCCs) are perfectly aligned with ryanodine receptors (RyR2) on the sarcoplasmic reticulum. This tight structural relationship is essential for a coordinated calcium release, allowing the heart to contract efficiently [12].

In contrast, heart failure with a reduced ejection fraction (HFrEF) disrupts this architecture. As the heart dilates and the walls thin, mechanical stress increases. Over time, this triggers structural remodeling, which damages the T-tubule system [5,12]. At the molecular level, proteins responsible for maintaining T-tubule integrity like junctophilin-2 (JPH2) and amphiphysin-2 (BIN1) become downregulated. Caveolin-3, another important structural protein, also shows signs of dysfunction in failing hearts [12]. Adding to this, the cytoskeleton becomes unstable, and changes in the microtubule network interfere with how LTCCs are delivered to and anchored in the cell membrane [12]. The result is a fragmented and disordered T-tubule network, often infiltrated by fibrous tissue. This leads to the separation of LTCCs and RyR2s, a condition referred to as “orphaned” RyR2s, which disrupts calcium signaling and weakens the force of contraction [5,12].

Heart failure with preserved ejection fraction (HFpEF), however, tells a different story. Instead of the dilated chambers seen in HFrEF, HFpEF typically involves concentric thickening of the heart wall. This remodeling helps to normalize wall stress and appears to protect the T-tubule system from the kind of damage seen in HFrEF [5,12]. In fact, evidence from both experimental models and patient biopsies shows that the T-tubule network in HFpEF is often preserved, and in some cases, even slightly enhanced. Adaptive responses, like increased tubule formation or slight dilation, may help maintain this integrity [5,12]. Because of this, calcium channels and receptors remain closely linked, supporting the normal release of calcium and preserving systolic function.

In summary, while HFrEF is marked by structural breakdown and misfiring calcium signals, HFpEF largely retains its architectural framework, allowing the heart to maintain its pumping ability. These differences underscore the fact that HFrEF and HFpEF are not simply points on the same spectrum, but distinct conditions with different underlying mechanisms [5].

Although LTCC function is preserved, HFpEF can be regarded as a disease of maladaptive signaling and diastolic Ca^2+^ management [2]. Of note in HFpEF, there is delayed relaxation as well as enhanced diastolic tension [21]. At the single-cell level, the diastolic Ca^2+^ content is elevated in HFpEF cardiomyocytes [4]. Kilfoil et al. noted that HFpEF rat myocytes had elevated intracellular calcium levels, which in turn increased the myocyte stiffness and resting tension [4]. The elevated diastolic Ca^2^⁺ levels were not due to severely impaired SR Ca^2^⁺ reuptake; in that study, the sarcoplasmic/endoplasmic reticulum calcium ATPase 2a( SERCA2a) protein levels and phospholamban (PLB) phosphorylation were not changed in HFpEF [4]. Instead, the authors suggest that increased Ca^2^⁺ influx through the LTCC and Ca^2^⁺ leakage from RyR2 channels may increase the diastolic Ca^2^⁺set point in HFpEF [4]. That is, Ca^2^⁺ release during systole is normal, but there is a mild Ca^2^⁺ overload during diastole as a result of a small increase in LTCC current and SR Ca^2^⁺ leak [4]. This pathway is regulated by Ca^2^⁺/calmodulin-dependent protein kinase II (CaMKII), which phosphorylates LTCCs and RyR2 to increase Ca^2^⁺ influx and diastolic SR Ca^2^⁺ leak [8]. The increased influx of Ca^2^⁺ into the SR increases the SR Ca^2^⁺ load, which together with RyR2 sensitization, leads to inappropriate Ca^2^⁺ release during diastole [8]. The RyR2-mediated leak is probably caused by increased phosphorylation at the Ser2808 site, a post-translational modification that increases the open probability of RyR2 without significantly depleting the SR Ca^2^⁺ stores [4]. The RyR2-mediated diastolic Ca^2^⁺ leak results in elevated resting cytosolic Ca^2^⁺ levels and activates Na⁺/Ca^2^⁺ exchanger inward currents that disrupt calcium homeostasis and membrane stability. The elevated diastolic Ca^2^⁺ and Na⁺/Ca^2^⁺ exchanger activity leads to delayed afterdepolarizations, which are low-amplitude depolarizations that can trigger arrhythmias when the heart is in a vulnerable state [8]. This finding is in agreement with other studies that have shown that HFpEF (especially in the presence of metabolic conditions like diabetes) is associated with reduced Ca^2^⁺ extrusion and elevated diastolic Ca^2^⁺ levels in myocytes [5]. The diastolic Ca^2^⁺ dysregulation in HFpEF leads to impaired relaxation and increased stiffness of the myocardium; bound troponin-Ca^2^⁺ increases the resting tension, and decreased titin phosphorylation (due to decreased protein kinase G (PKG) activity) increases the stiffness of cardiomyocytes [4]. Hence, HFpEF can be regarded as a disease of normal LTCC function during systole and disturbed diastolic Ca^2^⁺ handling.

Another aspect of LTCC regulation in HFpEF is blunted β-adrenergic responsiveness. Clinically, HFpEF patients have a poor exercise capacity and chronotropic/inotropic reserve. At the cellular level, studies indicate that HFpEF myocytes fail to augment Ca^2+^ cycling effectively upon β-adrenergic stimulation [4]. In the aforementioned rat model, isoproterenol (a β-agonist) elicited a much smaller increase in Ca^2+^ transient amplitude and SR Ca^2+^ reuptake in HFpEF cells versus the controls [4]. This suggests a desensitization of β-adrenergic signaling in HFpEF. Possible mechanisms include the downregulation or uncoupling of β_1-adrenergic receptors, the upregulation of inhibitory β_3 receptors, or impaired cAMP generation in HFpEF hearts [4,5]. The end result is that LTCC phosphorylation by protein kinase A under stress is subdued, limiting the augmenting effect on L-type Ca^2+^ current and Ca^2+^ reuptake. In effect, HFpEF hearts likely operate near their maximal Ca^2+^ utilization at the baseline (to preserve EF), leaving little reserve when adrenergic demand increases [4]. This “ceiling effect” contributes to exercise intolerance in HFpEF. It also partly explains why therapies effective in HFrEF (which reduce excessive β-stimulation) have not shown benefit in HFpEF as HFpEF already has muted β-adrenergic signaling [4].

Preserved LTCC expression and baseline function allows HFpEF to present a normal systolic Ca^2+^ release; this is the opposite of HFrEF, which displays LTCC deficiency [4,5]. HFpEF cells also have maladaptive Ca^2+^ handling and signaling in diastole, however, they have elevated resting Ca^2+^ and a reduced β-adrenergic reserve [4]. These molecular distinctions make it clear that HFpEF is not merely ‘mild HFrEF’ but rather a distinct entity more typically inspired by systemic inflammation, oxidative stress, and hypertrophic remodeling rather than the severe cellular Ca^2+^ depletion seen in HFrEF [5,22]. A concise side-by-side comparison of these phenotype-specific features is provided in Table 1.

### 3.3. Role of CaMKII in LTCC Regulation

Ca^2+^/calmodulin-dependent protein kinase II (CaMKII) has emerged as an important regulator of cardiac ion channels along with Ca^2+^. In the cardiac myocyte, CaMKII integrates signals of Ca^2+^ and oxidative stress [23]. Along with this, it can modify LTCC function via phosphorylation and other mechanisms [22]. Both HFrEF and HFpEF exhibit increased CaMKII activation, however, the consequences for LTCC regulation can differ.

Under physiological conditions, CaMKII is responsible for calibrating LTCC activity through a process known as Ca^2+^-dependent facilitation [1,2]. Repetitive Ca^2+^ influx leads to incremental increases in LTCC opening probability and current amplitude, partly mediated by the CaMKII phosphorylation of LTCC subunits [8,19]. The molecular targets include sites on the LTCC β subunit (e.g., Thr^498 on β_2a) and possibly the α_1C subunit, which, when phosphorylated by CaMKII, stabilizes the channel in a high-activity gating mode [8]. CaMKII-catalyzed phosphorylation promotes the mode-2 gating of LTCCs (a dysfunctional state of channel behavior characterized by prolonged openings and high open probability, leading to excessive Ca^2^⁺ influx)—characterized by longer openings and higher open probability [8]. This results in a larger Ca^2+^ influx for a given depolarization but also slowed inactivation of L-type Ca^2+^ current [8]. In disease settings, excessive CaMKII activity can lock LTCCs into this facilitated mode inappropriately. In fact, hyperactive LTCC gating due to CaMKII has been observed in failing hearts [20]. Mørk et al. reported that single LTCCs from failing human myocytes showed higher open probability and availability (consistent with CaMKII-mediated facilitation) despite a reduced total number of channels [11,19]. This CaMKII-driven enhancement of LTCC openings contributes to arrhythmogenic depolarizations: mode-2 gating and late Ca^2+^ currents can precipitate early afterdepolarizations and trigger ectopic beats [8]. Moreover, by increasing Ca^2+^ influx, CaMKII can lead to SR Ca^2+^ overload and subsequent diastolic Ca^2+^ leak via RyR2, promoting delayed afterdepolarizations [8].

CaMKII activity has been found to be elevated in heart failure (HF). Both HFpEF and HFrEF patients show elevated CaMKIIδ isoforms and increased autophosphorylation levels [22]. HFrEF patients experience chronic adrenergic stimulation and disrupted calcium homeostasis that lead to CaMKII overexpression, creating a negative feedback loop that damages calcium handling [8,22]. CaMKII maintains constitutive activity through autophosphorylation at Thr^287 or oxidative modification of Met^281/282 within its regulatory domain [8,22]. HF patients with comorbid conditions of diabetes and obesity show elevated reactive oxygen species (ROS) levels, which define oxidative stress as a characteristic feature of HF. The elevated ROS levels cause CaMKII oxidation at Met^281/282, resulting in prolonged enzyme activity even when the calcium levels normalize [8,22]. Pathological remodeling has been associated with the oxidized form of CaMKII since mice expressing oxidation-resistant CaMKII (Met281/282Val) develop protection against angiotensin II-induced hypertrophy and heart failure [22]. CaMKII functions as an essential intersection between β-adrenergic signals and redox stress to control LTCC activity [24]. Systemic inflammation and reactive oxygen species may activate CaMKII in HFpEF patients since β-adrenergic stimulation is less prominent than in HFrEF [22]. The data have established CaMKII as a key integrator between neurohormonal (β-adrenergic) and redox stress pathways that regulate LTCC activity.

The downstream impact of CaMKII hyperactivity includes LTCC dysfunction and Ca^2+^ mishandling [24]. In HFrEF, overactive CaMKII contributes to the reduced Ca^2+^ channel availability (through enhanced internalization or altered channel turnover) but also paradoxically to increased late Ca^2+^ currents and arrhythmia risk via mode-2 gating [8,20]. CaMKII also phosphorylates RyR2, exacerbating SR Ca^2+^ leak in HF, and phosphorylates phospholamban (PLB) at Thr^17, which can alter SERCA function [22]. In HFpEF, CaMKII may play a role in the blunted β-adrenergic reserve: chronic CaMKII activation can induce the desensitization of β-receptor signaling and contribute to altered phosphodiesterase activity, limiting the effects of cAMP/protein kinase A on LTCCs [22,23]. There is also evidence that CaMKIIδ translocates to the nucleus and alters transcription in HF, sustaining a program of fetal gene expression and hypertrophy that could indirectly modulate LTCC subunit expression or T-tubule structure [22].

In failing human hearts, the expression of CaMKIIδ is upregulated, particularly the CaMKIIδ_B splice variant, which contains a nuclear localization signal that directs it to the cardiomyocyte nucleus [25]. Once translocated, nuclear CaMKIIδ_B initiates maladaptive transcriptional programs associated with cardiac hypertrophy and progression to heart failure. A key mechanism involves phosphorylation of class IIa histone deacetylases (HDAC4 and HDAC5), which are subsequently exported from the nucleus. This relieves their inhibition of myocyte enhancer factor 2 (MEF2), a transcription factor that promotes the expression of hypertrophic genes [25].

This calcium sensitive signaling cascade leads to the reactivation of a fetal gene expression profile, marked by the upregulation of atrial natriuretic factor (Anf), B type natriuretic peptide (Bnp), β myosin heavy chain, and skeletal α actin [25]. In addition to modulating hypertrophic gene programs, sustained CaMKII activity affects calcium handling proteins. Prolonged activation has been associated with the downregulation of SERCA2a and phospholamban (PLB), along with the increased expression of the Na⁺/Ca^2^⁺ exchanger (NCX1) changes that contribute to intracellular calcium overload and diastolic dysfunction [26].

Together, these findings illustrate that CaMKIIδ_B not only regulates ion channel function, but also drives transcriptional reprogramming in heart failure, linking electrical activity, stress signals, and long-term gene expression changes.

What is clear is that CaMKII is a major “villain” in HF pathophysiology [8,23]. Its broad range of targets—LTCC, RyR2, PLB, etc.—means that CaMKII hyperactivity can simultaneously weaken contractility and increase arrhythmogenic risk [8,24]. For instance, CaMKII-dependent LTCC phosphorylation and RyR2 phosphorylation have been shown to synergistically promote early afterdepolarizations in HF models [20]. The recognition of CaMKII’s centrality has made it an attractive therapeutic target (discussed below). Ongoing research is also exploring how to specifically interrupt CaMKII’s interaction with LTCC complexes—for example, preventing CaMKII docking on the LTCC β_2 subunit or inhibiting its oxidation—as a means by which to normalize LTCC behavior in failing hearts [8,20].

CaMKII acts as a critical modifier of LTCC function in HF. It becomes persistently active in both HFrEF and HFpEF, leading to LTCC hyperphosphorylation. This results in increased LTCC open probabilities and slowed inactivation (facilitation) [8], which can aggravate Ca^2+^ dysregulation—contributing to arrhythmias (by triggering aberrant depolarizations) and contractile dysfunction (by depleting SR Ca^2+^ or impairing relaxation) [4,8,15]. Targeting CaMKII and its downstream effects on LTCCs thus represents a logical strategy to improve cardiac function in HF.

### 3.4. Emerging Therapeutic Targets

Given the pivotal role of LTCC dysfunction and CaMKII in HF pathogenesis, several emerging therapeutic strategies aim to normalize LTCC regulation or its downstream consequences. Traditional HF therapies (β-blockers, renin–angiotensin system [RAS] inhibitors) provide indirect benefits by reducing chronic adrenergic and hemodynamic stress, but newer approaches seek a more direct modulation of the cardiac Ca^2+^ handling machinery.

#### 3.4.1. CaMKII Inhibition

As CaMKII is a molecular lynchpin exacerbating both systolic and diastolic Ca^2+^ defects, it has become a prime target for drug development [8,22]. Preclinical studies using genetic CaMKII inhibition or peptide inhibitors (like autocamtide inhibitory peptide (AIP) and CaMKII inhibitor peptide (CaMKIIN) have shown improved cardiac function and reduced arrhythmias in HF models [22]. Until recently, CaMKII was considered ‘undruggable’ due to its ubiquitous roles, but new small-molecule inhibitors such as autocamtide inhibitory peptide (AIP), CaMKIIN, AS105, and hesperadin have emerged, showing selective efficacy in cardiac models [27,28]. In experimental HFpEF models characterized by high oxidative stress, the inhibition of CaMKII oxidation, achieved via the overexpression of methionine sulfoxide reductase or through the mutation or CaMKII’s Met^281^/^282^, helped to preserve diastolic function. This highlights the therapeutic potential of targeting CaMKII redox activation [22]. CaMKII inhibitors, such as autocamtide inhibitory peptide (AIP), CaMKIIN, AS105, and hesperidin, could provide a dual benefit for heart failure by enhancing contractility through the stabilization of Ca^2^⁺ cycling and by reducing arrhythmic risk.This would be particularly advantageous for HFpEF patients, who currently lack effective treatment options, through the direct targeting of Ca^2^⁺ dysregulation driven by inflammation and CaMKII activation—mechanisms not effectively addressed by conventional heart failure therapies or non-specific anti-inflammatory agents [29,30].

#### 3.4.2. LTCC Modulation

Direct modulation of the activity of LTCC constitutes another strategy. The use of dihydropyridines, such as amlodipine, represents traditional LTCC blockers that serve as useful antihypertensive and antianginal medications but demonstrate an unfavorable impact on systolic function when used in HFrEF by additionally reducing Ca^2+^ entry. There has been increased interest in developing more precise LTCC modulators. One method is to target accessory proteins that regulate LTCC trafficking and gating. For instance, the small GTPase Rad normally limits LTCCs, and its levels are reduced in HFrEF, causing the remaining channels to have an increased open probability [20]. Strategies aimed at the restoration of Rad function may assist in the modulation of LTCC activity without affecting channel expression. Another potential strategy is to focus on particular LTCC subtypes, for instance, Cav1.3, which is a minor L-type channel isoform that is re-expressed in the failing ventricles and may give rise to arrhythmogenic slow Ca^2+^ currents [11]. Selective Cav1.3 inhibition, though still in the preclinical stages, may offer a future strategy to prevent arrhythmias in HF without compromising Cav1.2-dependent contractility [31]. Additionally, insights from natural toxins such as maitotoxin and atrotoxin, which activate calcium channels, offer valuable models for the design of LTCC agonists. While these toxins are not therapeutically viable due to their high toxicity, they help identify potential gating sites and conformational changes that could inspire future agonist drug development [32,33,34,35,36].

#### 3.4.3. RyR2 Stabilizers and Ca^2+^ Cycling Enhancers

Although targeting the SRCa^2+^ release channel is one step downstream of LTCC, it directly addresses the coupling between LTCC trigger and SR Ca^2+^ output. Compounds known as Rycals (e.g., ARM210) have been designed to stabilize RyR2 in the closed state, reducing diastolic Ca^2+^ leak [2,3]. By fixing the “leaky valve” of the SR, these drugs can improve the SR Ca^2+^ content and contractility while lowering arrhythmogenic Ca^2+^ sparks. A RyR2 stabilizer (e.g., elamipretide) could complement LTCC normalization—the LTCC provides the trigger, and a healthier SR provides a stronger, non-leaky Ca^2+^ response. Some Rycals have reached clinical trials for HF or catecholaminergic polymorphic VT [3]. Similarly, enhancing SERCA2a activity via gene therapy (Ad-SERCA2a, tested in the CUPID trial) was an attempt to improve Ca^2+^ reuptake and contractile reserve in HFrEF [3]. While the initial SERCA gene therapy results in humans were modest, ongoing refinements (like targeted delivery or combined modulation of PLB) are in progress. These approaches, though not directly acting on LTCC, influence the same EC coupling axis and thus are relevant to overall Ca^2+^ homeostasis in HF.

## 4. Future Directions

The treatment of HFrEF could go beyond the current standard practice by developing drugs that activate LTCC current either only during systole or at rates without diastolic activation to increase contractility with less resting leaks. Gene therapy or RNA-based approaches that enhance BIN1 or junctophilin-2 expression would create more efficient LTCC–RyR coupling within failing hearts. Research should focus on identifying facilitators of LTCC dysfunction because CaMKII inhibitors will require clinical trials as a first step toward creating new HF treatments based on molecular substrates of arrhythmias. The Cav1.3 LTCC subunit, along with other LTCC subunits present in HF, show potential for future investigation to develop specialized regulators that could fine-tune cardiac contractility and excitability. HFpEF research demands urgent investigation into how metabolic and inflammatory factors shape cardiomyocyte calcium regulation mechanisms. Large animal HFpEF models, together with iPCS-derived cardiomyocytes from HFpEF patients, enable scientists to study how comorbidities affect LTCC function and Ca^2+^ cycling mechanisms. The medical community shows growing interest in developing treatments for specific HFpEF profiles [21]. Anti-cytokine or antioxidant therapy could treat an inflammation-mediated HFpEF by normalizing CaMKII and LTCC function, but HFpEF patients with hypertension as the primary cause would benefit from phenotype-specific Ca^2+^-handing modulators or myosin light chain kinase (MLCK) inhibitors to enhance relaxation [21]. The combination of mechanical and molecular therapies offers a new direction, as exercise conditioning and innovative filling devices used together with pharmacologic agents like sodium-glucose cotransporter-2 (SGLT2) inhibitors or guanylate cyclase agonists would create synergistic benefits for Ca^2+^ regulation and diastolic function. Omics and system biology approaches will help identify new regulatory elements of LTCC function through microRNAs and scaffolding proteins that change expression during HF. Treatment of these upstream regulators presents an opportunity to fix the initial Ca^2+^ handling dysfunction. Recent scientific findings show that microRNAs repress LTCC subunits in HF patients through repression mechanisms that could be used to modify LTCC expression in HFrEF and reduce excessive Ca^2+^ influx in HFpEF.

The implementation of therapeutic treatments based on molecular findings requires precise adjustments to avoid disturbing the sensitive Ca^2+^ equilibrium needed for heartbeat maintenance. The future treatment of HF will require precise medical approaches to establish which patients have Ca^2+^ deficiency or Ca^2+^ management issues so that physicians can provide appropriate treatments. Medical professionals can develop personalized cardiac treatments by studying HFrEF and HFpEF at the molecular level including LTCC regulation and its subsequent effects. The effectiveness of treatments for HFrEF versus HFpEF will be established through ongoing and upcoming clinical trials examining the CaMKII and LTCC microdomains and diastolic Ca^2+^ management. Research into LTCC regulation in HF shows potential to develop better treatments for the distinct cellular abnormalities found in HFrEF and HFpEF.

In line with emerging research, future therapeutic strategies for HF are increasingly focusing on the precision targeting of dysregulated Ca^2^⁺ handling and CaMKII activity. For example, highly selective CaMKII inhibitors are now in development: Cardurion’s first-in-class molecules (e.g., CRD-2015, CRD-2959) demonstrated a potent inhibition of CaMKII targets and improved survival in preclinical HF models [37], and a lead agent has already entered Phase 1 trials (with plans to expand into HF indications) [37]. Efforts are also turning to microdomain specific interventions that stabilize local Ca^2^⁺ signaling complexes; notably, gene delivery of the T-tubule organizer cBIN1 restored dyadic microdomains and reversed heart failure phenotypes in animal models [38]. Recognizing CaMKII’s pathologic redox activation in HF, new redox-sensitive approaches are being explored including precision gene editing to eliminate CaMKIIδ’s oxidation susceptible methionines [39] or even its autophosphorylation site [40], which has yielded cardioprotective effects in recent preclinical studies.

Next-generation gene therapies targeting calcium handling are gaining momentum: for instance, an AAV1 (adeno-associated virus serotype 1) -mediated SERCA2a gene transfer was recently tested in HFpEF patients (first in human), showing no major safety issues and early improvements in exercise capacity and cardiac biomarkers [41]. Similarly, CRISPR/Cas9 base editing of pathogenic calcium regulators (e.g., CaMKIIδ) has proven feasible in cardiomyocytes and mice, suggesting a permanent way to blunt maladaptive Ca^2^⁺ signaling [39]. Meanwhile, RNA-based therapeutics are under active investigation to fine-tune myocardial Ca^2^⁺ homeostasis, for example, antisense inhibition of microRNA-132 (CDR132L) was safe in Phase 1 and showed encouraging improvements in cardiac function in HF patients [42]. Finally, patient-specific iPSC-derived cardiomyocytes have emerged as platforms for drug testing and personalized HF modeling, enabling the high-throughput screening of therapies on patient matched cardiac cells [43]. These forward-looking approaches, now accelerating from the bench to clinic, underscore a more hopeful therapeutic horizon targeting CaMKII and calcium dysregulation in heart failure.

## 5. Conclusions

The two distinct paradigms of LTCC dysregulation present as heart failure with reduced EF and preserved EF. The combination of chronic stress results in HFrEF patients losing LTCC density while their LTCC–RyR coupling becomes disrupted, which leads to weakened EC coupling and systolic failure. The excessive kinase activation of PKA and CaMKII in HFrEF leads to the hyperphosphorylation of LTCCs and RyR2, which produces Ca^2+^ leakage and arrhythmias. The heart maintains its systolic output through compensation in HFpEF, even though LTCC expression and baseline function remain mostly intact. The myocytes of HFpEF patients experience maladaptive signaling through elevated diastolic Ca^2+^ levels and myocyte stiffening and reduced β-adrenergic response, which impairs relaxation and reserve. The enzyme CaMKII functions as a common link between pathological LTCC modifications that cause both LTCC dysfunction in HFrEF and elevated diastolic Ca^2+^ levels in HFpEF. The distinct mechanisms of disease explain why HFrEF treatments that decrease sympathetic drive and reverse remodeling have not led to better results in HFpEF patients.

## Figures and Tables

**Figure 1 ijms-26-05738-f001:**
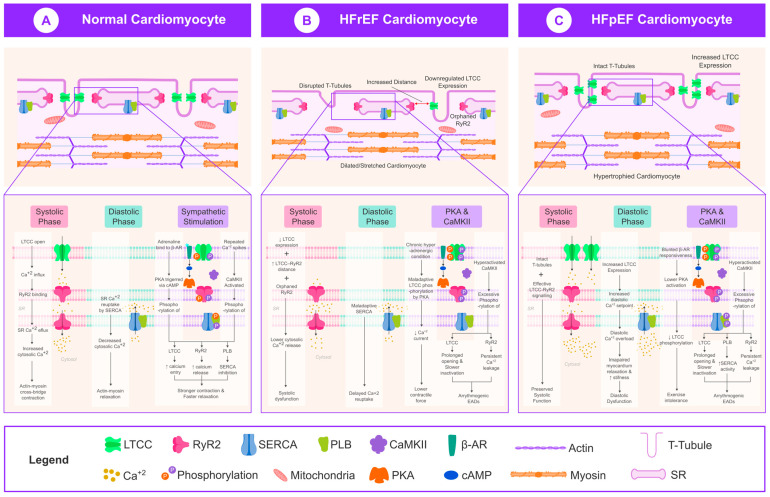
Schematic representation of L-type calcium channel (LTCC) behavior in normal, HFrEF, and HFpEF cardiomyocytes. (**A**) Normal cardiomyocyte showing organized T-tubules and tightly coupled LTCC–RyR2 interactions for efficient systolic Ca^2^⁺ release and diastolic reuptake. (**B**) HFrEF cardiomyocyte with disrupted T-tubules, increased LTCC–RyR2 distance, downregulated LTCC expression, maladaptive phosphorylation, and delayed Ca^2^⁺ reuptake—leading to systolic dysfunction and arrhythmogenesis. (**C**) HFpEF cardiomyocyte with preserved or increased LTCC expression, intact T-tubules, elevated diastolic Ca^2^⁺, and blunted β-adrenergic responsiveness—leading to preserved EF but impaired relaxation. The figure illustrates phase-specific Ca^2^⁺ flux, LTCC gating, kinase signaling, and EC coupling efficiency across phenotypes. Abbreviations: LTCC, L-type calcium channel; RyR2, ryanodine receptor 2; SERCA, sarcoplasmic reticulum Ca^2^⁺-ATPase; PLB, phospholamban; CaMKII, calcium/calmodulin-dependent protein kinase II; β-AR, beta-adrenergic receptor; PKA, protein kinase A; cAMP, cyclic AMP; HFrEF, heart failure with reduced ejection fraction; HFpEF, heart failure with preserved ejection fraction; EADs, early afterdepolarizations.

**Table 1 ijms-26-05738-t001:** Summary of the key molecular, structural, and functional contrasts in LTCC-mediated Ca^2+^ handling between HFrEF and HFpEF cardiomyocytes.

Feature	HFrEF	HFpEF
LTCC expression and peak L-type Ca^2^⁺ current	↓ (~30–40% loss); diminished channel binding capacity	Normal or slightly ↑
T-tubule architecture	Sparse, disorganized network → “orphaned” RyR2s	Preserved or mildly ↑ density
Post-translational state of LTCC/RyR2	Chronic PKA + CaMKII hyper-phosphorylation → maladaptive gating	Limited PKA activity; CaMKII phosphorylation drives diastolic Ca^2+^ leak
LTCC–RyR2 coupling distance	Widened → lower excitation–contraction gain	Normal/narrow → efficient systolic trigger
Systolic Ca^2+^ release and contractility	Depressed Ca^2+^ transients → systolic dysfunction	Preserved Ca^2+^ transients → maintained EF
Diastolic Ca^2+^ handling	Normal-to-low resting Ca^2+^	Elevated diastolic Ca^2+^, higher passive tension
β-adrenergic reserve	Hyper-adrenergic environment yet desensitized receptors	Blunted reserve (“ceiling effect”)
Predominant arrhythmogenic trigger	Early after-depolarizations (late Ca^2^⁺ entry + RyR2 leak)	Delayed after-depolarizations (diastolic Ca^2^⁺ overload)

Symbols: ↑ = increased; ↓ = decreased; → = leads to/resulting in.

## Data Availability

All of the data reported are available on the PubMed, and Google Scholar web databases.

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
