# Peer review of "Molecular Mechanisms of L-Type Calcium Channel Dysregulation in Heart Failure"

_ijms, 2025, doi:10.3390/ijms26125738_

Round 1

Reviewer 1 Report

Comments and Suggestions for Authors

Reviewing report

The manuscript submitted by ARBAB Khalid and collaborators entitled “Molecular Mechanisms of L-Type Calcium Channel Dysregulation in Heart Failure” is a review about the knowledges from the last decade of the role of L-Type calcium channel in heart failure and more specifically with a comparison between reduced and preserved ejection fraction. In this review, ARBAB Khalid and collaborators discuss in the first time about the dysregulation of L-type channel in heart failure with reduced ejection fraction compared to preserved ejection fraction. ARBAB Khalid and collaborators described the differences reported in the mechanisms of the calcium channel dysregulation in the two pathophysiological cases. Authors, give schematically representation of the different mechanisms involved and a table with comparisons of observed characteristics between heart failure with reduced ejection fraction and heart failure with preserved ejection fraction. In the second part, ARBAB Khalid and collaborators discuss about the role of calcium calmodulin dependent protein kinase II CAMKII in the regulation and dysregulation of L-type calcium channel with different mechanisms between heart failure with reduced ejection fraction and heart failure with preserved ejection fraction. In the last part of the review, authors discuss about emerging therapeutic targets including CAMKII inhibition, L-type calcium channel modulation and ryanodine receptors. The present review does the state of art of heart failure with reduced or preserved ejection fraction from the last decade and is interesting to have an idea about the present knowledges in heart failure physiopathology. However, in my opinion, it misses several major and minor modifications to make better the present review for the publication.

Major comments:

  • Figure 1 is a schematically representation of heart failure with reduced ejection fraction. In my opinion the picture is really basic and not really shows and explains the dysregulation with mechanisms in details. Could authors present a better picture to illustrate their manuscript? Moreover, it could be better and strong if this picture would be compared to another one representing “normal or physiological condition” …

  • In their 3.2 part of the review authors discuss about the L-type calcium channel dysregulation in heart failure with reduced ejection fraction. This section is not really completed and more information and references could be added. For example, explain why in heart failure with reduced ejection fraction the tubules t structure is disrupted and why is not the case in heart failure with preserved ejection fraction. What are the mechanisms that could explain these modifications of the tubule t structure and what are about these mechanisms in the two pathophysiological conditions compared to physiological condition?

  • In their manuscript authors discuss about the putative role of CAMKII on genes expression after its translocation in the nucleus. Could authors develop this part giving different gens and references?

  • In their manuscript authors discuss about the emerging therapeutic targets and the future direction in order to treat heart failure. In my opinion, this part is not really relevant if it is not better. Indeed, information, given by authors are not really new and informative for the putative therapeutic alternatives with new drug and targets. Could authors develop more this part of the manuscript with a better vision of the future in the context pathophysiological?

Reviewer 2 Report

Comments and Suggestions for Authors

Reference:  “Molecular Mechanisms of L-Type Calcium Channel Dysregulation in Heart Failure Arbab Khalid et al., 2025 IJMS.”

General comments: In the manuscript submitted for publication in IJMS, the authors are reporting their results from analyses of data previously published in the literature through searches in the PubMed and Google Scholar databases between the years 2015–2025, relating the role of L-type calcium channels (LTCCs) with myocardial alterations and suggesting LTCC dysregulation as a mechanism that causes myocardial dysfunction. According to the authors, the findings can provide clinicians and researchers with a comprehensive reference for current concepts and future therapeutic approaches in heart failure. After reading this review text, I believe that it is a text with potential for publication in the IJMS; however, I would like to indicate suggestions that could make a revised version of the text more complete from a scientific point of view and attractive to readers. 

Specific Comments:

1-  Between lines 31 to 35 (key words) I suggest that the authors make the text more concise and summarized, after all we have key words and not an abstract. How about L-type calcium channel; Ca²⁺/calmodulin-dependent protein kinase II; heart failure; calcium handling. 

2- In my opinion the text written between lines 53 and 54 Deserves reference citation …. Notably, HFrEF and HFpEF exhibit disparate remodeling processes and responses to treatments, suggesting that they have different molecular mechanisms.  

3- In the line 57 authors wrote … This review provides a summary of recent literature (from the last ten years). I suggest change the sentence to … This review was based on literature searches in the area using PubMed (https://pubmed.ncbi.nlm.nih.gov) and Google Scholar (https://scholar.google.com) databases about published articles between years 2015–2025.  

4- In the line 63 … emerging therapeutic strategies targeting these pathways. Here, authors could indicate the types of drugs potentially used, such as inhibitors, agonists, competitors, among others. Leaving an indication of the families of drugs hypothetically indicated. This could increase reader appeal. It could also be indicated in the abstract, since the abstract text is generally the text included in literature search sites. 

5- In the line 66 authors wrote ….A literature review was performed using PubMed, and Google Scholar databases. Here authors could write the complete electronic address of Sites used in research.  

6-  In the line 69 authors wrote…the date range was from 2015 to 2025. I suppose a ten year interval for research is a good and reliable time, but here authors could provide some explanation, as in the end only 17 publications were used, which is not a large number. 

7- Among lines 65 to 71 authors could include that only 17 published articles were finally used to obtain the conclusions.  

8- The text written in the lines   75 and 76 …. Heart failure with reduced ejection fraction is classically associated with weakened EC coupling and depressed systolic Ca2+ transients.    Could be accompanied by references.  

9- In the lines 86 an 87 authors wrote …Along with a reduced number of channels, HFrEF is characterized by post-translational modifications of LTCC which alter channel gating [12].  Since the term post-transcriptional modification is very broad, here the authors could indicate which modifications are involved with the phenomenon. For example, N- or O-glycosylations, Disulfide bridge formations, Proteolytic cleavages, among other possibilities, besides phosphorylation pointed in the example. 

10- About figure 1. In the revised versition of figure authors must complete information of figure in the legend. Such as: 1- Even that abbreviations were indicated along the text, they could be informed in the legend. 2- There are structures in the figure that were not indicated in the legend. Are orange-colored elliptical structures mitochondria? 3- What are fibrilar structures colored in purple? 4- What are structures colored in orange and connected to the purple structures by light green lines? What are the light green lines? 5- What structures containing RyR2, SERCA, PLB receptors .are Smooth Endoplasmic Reticulum?   As pointed several parts of figure must be defined!!! 

11- If the authors really want to emphasize the role of calcium channel phosphorylation in pathophysiological events, I suggest that they add to Figure 1 an insert showing molecular aspects of the channels. As shown, it is very generic.  

12- In my opinion, the authors could also indicate in the legend a flowchart with the apparent order of events, to facilitate readers' understanding. 

13- My comments indicated for figure 1 are also indicated for the legend of figure 2. Authors could indicate in the caption the meaning of the abbreviations, indicate a flowchart of events, and indicate the names of all the structures shown in the figure. 

14- It seems to me that throughout the text the authors make a comparative analysis between reduced ejection fraction (HFrEF) and heart failure with preserved ejection fraction (HFpEF), with data shown in figures 1 and 2 respectively for (HFrEF) and (HFpEF). In order to understand the pathophysiology, it is interesting to show a more physiological model first. Thus, to facilitate readers' understanding, the authors could publish a figure that would be Figure 1 in the revised manuscript, which would represent a cell under normal conditions, without the changes described in the two pathophysiological conditions described. It seems to make more sense to me. I would like to know the opinion of authors.  

15- I am confused about the figures shown in the text. The authors need to make it clear that the figures shown in this publication are not the same as other figures already previously published, which would constitute self- plagiarism. 

16- For the readers' understanding, wouldn't it be interesting if Table 1 came right after item 3.3 and not right below figure 2 as it appears in the text? 

17- In the lines 310 to 312 ….Until recently, CaMKII was considered “undruggable” due to its ubiquitous roles, but new small-molecule inhibitors are under investigation.  In my opinion authors could indicate some references for this sentence.  

18- In the lines 316 and 317 authors wrote …CaMKII inhibitors could provide a dual benefit for heart failure. For the manuscript to gain importance, the authors could indicate in the revised version the names of some inhibitors already described. 

19- In the lines 318 to 321 authors wrote …..This would be particularly advantageous for HFpEF patients, who currently lack effective treatment options, through the direct targeting of  dysregulation driven by inflammation and CaMKII activation, which is not effectively addressed by conventional heart failure therapies.   Wouldn't conventional anti-inflammatories be good models of therapies for the indicated conditions? 

20- In the lines 336 and 337 …Inhibition of Cav1.3 selectively may be useful to prevent arrhythmias in HF without affecting Cav1.2 function and hence contractility. But do such inhibitors already exist to the point where one can hypothesize such activities?  

21- What to expect from toxins that activate calcium channel openings such as maitotoxin and atrotoxin? They could be good examples of work, looking for agonist drugs.  

Reviewer 3 Report

Comments and Suggestions for Authors

This is a well-written and comprehensive review that provides a mechanistically rich comparison between L-type calcium channel (LTCC) regulation in HFrEF and HFpEF. The topic is timely and clinically relevant, with strong support from contemporary literature. 
-"Facilitation" (used for CaMKII-mediated LTCC activity) and “mode-2 gating” are technical terms that may not be universally understood. A brief definition or reference in-text would aid readers.
-Line 71: Double punctuation: remove extra period after “meta-analyses..”
-Line 225: Table 1 caption: remove second period after "cardiomyocytes.."

Round 2

Reviewer 1 Report

Comments and Suggestions for Authors

The manuscript is OK for me with this form after some corrections